# Effect of Co and Cr on the Stability of Strengthening Phases in Nickelbase Superalloys

**Martin Bäker \*** and **Joachim Rösler**

Institute for Materials Science, Technische Universität Braunschweig, Langer Kamp 8,
38106 Braunschweig, Germany; j.roesler@tu-braunschweig.de
\* Correspondence: martin.baeker@tu-braunschweig.de; Tel.: +49-531-391-3065

**Abstract:** Nickel-base superalloys such as VDM 780 may possess a high content of Cr and Co. This influences solution energies of phase-forming elements such as Al and Ta ($\gamma'$-phase), Nb ($\gamma''$- and $\delta$-phase), and Ti ($\eta$-phase). We perform density functional theory studies of a nickel matrix at 0 K with high concentrations of either Co and Cr and calculate the influence of these elements on solution energies. In the case of Co, the solution energy can be predicted well by the nearest-neighbor interaction in the Co-rich matrix. For Cr, the effect is more complicated because Cr has a larger ionic radius and changes the magnetic state of the material. The effect of a Cr-rich matrix on the energy of Co is dominated by magnetic effects and interactions with the other elements by elastic deformation of the lattice. A high content of Co or Cr will thus increase the solvus temperature of the strengthening phase in nickel-base superalloys, in agreement with the literature and thermodynamic calculations.

**Keywords:** nickel-base alloys; phase stability; density functional theory





## 1. Introduction

Nickel-base superalloys possess an outstanding mechanical strength even at high temperatures due to the formation of the strengthening $\gamma'$-(Ni$_3$Al) and $\gamma''$-phase (Ni$_3$Nb). Developing new alloys with improved behavior is still an active field of research, both for single-crystalline materials [1,2] and for wrought alloys such as VDM Alloy 780 [3]. In wrought alloys, in addition to the strengthening phases, the $\delta$-(Ni$_3$Nb) and $\eta$-phase (Ni$_3$Ti) are important because they can be utilized to ensure a fine-grained structure. However, they may also form precipitations that embrittle the alloy during service [4].

To develop improved alloys, a detailed understanding of the precipitation behavior and stability of strengthening phases is needed. Modern alloy development approaches combine CALPHAD methods with experimental results and density functional theory, see for example [2]. Density functional theory can offer insights into the stability and the interfacial behavior of relevant phases [5–10].

To calculate the phase stability in density functional theory, a reference energy is needed. Usually, this is taken either as the energy of the pure materials (e.g., Al in the case of $\gamma'$) or as the solution energy of the element under consideration in a nickel matrix. However, in a wrought alloy such as VDM Alloy 780 [3], the matrix may contain a high amount of Co and Cr. The relevant energy difference for a phase-forming element such as Al is thus not the energy of this element in a pure nickel matrix, but the energy in the matrix containing other alloying elements.

To study the effect of alloying elements on phase stability, we perform density functional theory studies of the energy of the elements Al, Ti, Nb, and Ta in a nickel matrix with realistic concentrations of Co or Cr. In addition, we also study the case of Co in a Cr-rich matrix.



Both a Co-rich and Cr-rich matrix increase the energy of the phase-forming elements. The effect of this on the phase stability is discussed and compared to experimental results and thermodynamical calculations.

We also show that the solution energy can be understood simply by considering nearest-neighbor effects in a Co-rich matrix, whereas magnetic effects are important in a Cr-rich matrix.

## 2. Materials and Methods

All density functional theory calculations were performed using VASP with the PAW-PBE potentials [11–15]. For each element, the pseudopotential with maximum number of electrons was chosen. An energy cutoff of 520 eV and a first-order Methfessel–Paxton scheme with a smearing parameter of 0.07 eV were used. To ensure high precision, the precision parameter was set to "accurate", and real space projection operators were calculated to a precision of $10^{-4}$. The convergence criterion for the electronic loop was set to $10^{-4}$ meV, and the ionic loop during relaxation was stopped when the energy change was below $10^{-2}$ meV.

To calculate the correct volume of each supercell, the following procedure was used: In the first step, calculations with relaxation of the ionic positions and the cell shape were performed at fixed lattice volume for five different volumes. A Birch–Murnaghan state equation [16] was used to fit the energies and determine the optimum scale factor. For this scale factor, a final calculation was then performed with a *k*-point spacing of $0.1 \, \text{Å}^{-1}$. Some intermediate calculations were performed using the simpler method of directly relaxing the cell shape, cell size and the ions.

Except for some auxilliary calculations described below, all calculations were spin-polarized. Ni atoms were usually initialized with a moment of 1 $\mu_B$, Co atom with a larger value. For cells containing Cr, different initializations of the magnetic moment of the Cr atom were tried because Cr can be anti-ferromagnetic in a nickel matrix. Details are described in the results section.

Calculations of the solution and nearest-neighbor interactions were performed using supercells with 108 atoms. For calculations with a high Co content, 64-atom special quasirandom structure (SQS) cells were created using the software ATAT [17]. For the Cr-rich matrix, calculations with 32-atom SQS cells were performed (the smaller size was chosen because converging to the correct magnetic state requires a large number of trial runs). In addition, cells with composition $Ni_{26}Cr_6$ and $Ni_{25}Cr_6X$ that contained no nearest-neighbor bonds between Cr atoms were created using the software tool SOD (Site-Occupation Disorder) [18].

## 3. Results

### 3.1. Influence of Co

#### 3.1.1. Interaction Effects at Low Concentrations

The first step in creating supercells with a high content of an alloying element such as Co is to calculate the interaction energy of the element with itself when situated close in the lattice. If the energy of two alloying atoms increases strongly in a nearest-neighbor position, such configurations will not occur. On the other hand, attractive forces, for example on next-nearest neighbor positions, might lead to local ordering of the atoms.

The interaction energy $E_{\text{int, XY}}$ between two atoms X and Y in a certain configuration (nearest neighbor, next nearest neighbor, etc.) is calculated by comparing supercells containing the atoms in this configuration with two supercells with isolated atoms. Using 108 atoms in the supercell, the interaction energy is

$$E_{\text{int,XY}} = E(Ni_{106}XY) + E(Ni_{108}) - (E(Ni_{107}X) + E(Ni_{107}Y)). \tag{1}$$

Here, $E(\cdot)$ denotes the energy of the supercell. A negative interaction energy means that the configuration is energetically favorable, and a positive energy demonstrates that separating the atoms is more favorable.

Using Equation (1), the interaction energy between Co and Co and between Co and the phase-forming elements (Al, Ti, Nb, Ta) can be calculated. Results are shown in Table 1. The self-interaction of Co with itself is rather small, showing that Ni-Co bonds and Ni-Ni bonds have practically the same energy. Since the ionic radius of Co is almost identical to that of Ni, elastic interactions are negligible.

Placing one of the phase-forming elements next to Co has an energy cost of roughly 100 meV. There is no clear effect of the ionic radius of the elements because the lattice distortion is unchanged when one Ni neighbor is replaced by Co. Next-nearest-neighbor energies are generally negligible.

**Table 1.** Interaction energy between Co and other alloying atoms calculated in a matrix with 108 atoms. Energies shown are for nearest neighbor and next-nearest-neighbor positions.

| Element | NEAREST Neighbor | Next Nearest Neighbor |
|---------|------------------|-----------------------|
| Al | 92.2 meV | 0.0 meV |
| Ti | 103.0 meV | 2.7 meV |
| Co | 1.3 meV | −3.5 meV |
| Nb | 113.9 meV | 3.7 meV |
| Ta | 117.1 meV | 1.5 meV |

3.1.2. Interaction in Co-Rich Cells

Since there is no strong self-interaction of Co, a Co-rich nickel matrix will be disordered so that arbitrary configurations can be considered.

We created three special quasirandom structures, shown in Figure 1, using the program suite ATAT [17], with a composition of $Ni_{47}Co_{16}X$. To study the influence of high Co concentrations on the solution energy of an alloying element, we compare the energy of this cell with that of a cell where X is replaced with Ni. From this, the energy to transfer an element from a pure nickel matrix to a matrix containing 25 at-% Co can be calculated as follows:

$$E_{\text{Co-rich,X}} = E(Ni_{108}) + E(Ni_{47}Co_{16}X) - (E(Ni_{107}X) + E(Ni_{48}Co_{16})). \tag{2}$$

If this number is positive, placing the atom in a Co-rich matrix is less favorable than in a pure Ni matrix, so Co will tend to displace the element to other phases.

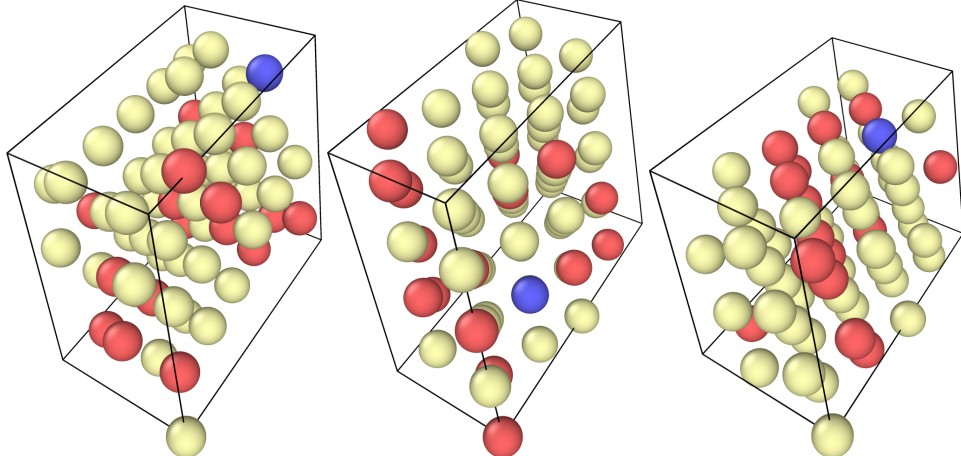

**Figure 1.** The 3 special quasirandom structures with composition $Ni_{47}Co_{16}X$; Ni atoms in yellow, Co atoms in red, alloying element in blue.

The results in Table 2 show that the energy is positive in all cases; thus placing an alloying element in a Co-rich matrix is unfavorable compared to a pure Ni matrix. The alloying atom has three Co-neighbors in the special quasirandom cell. We can account for this by adding three times the nearest-neighbor interaction energy to the second term

$$
\begin{aligned}
E_{\text{Co-rich},X,\text{nn}} &= E(\text{Ni}_{108}) + E(\text{Ni}_{47}\text{Co}_{16}X) - (E(\text{Ni}_{107}X) + E(\text{Ni}_{48}\text{Co}_{16}) + 3E_{\text{nn}}) \\
&= E_{\text{Co-rich},X} - 3E_{\text{nn}} \,.
\end{aligned}
\tag{3}
$$

$E_{\text{Co-rich},X,\text{nn}}$ thus quantifies the part of the transfer energy that is not explained by the nearest-neighbor interaction. As can be seen from Table 2, this energy is small in most cases, the largest deviation being $-56.9$ meV. This shows that the energies in the Co-rich matrix can be predicted quite well by the nearest-neighbor interaction.

**Table 2.** Energy $E_{\text{Co-rich}}$ to transfer an element from a pure nickel matrix to a matrix containing 25 at-% Co. Additionally shown is the energy $E_{\text{Co-rich,nn}}$ that accounts for nearest-neighbor effects. If this number is small, the effect of a Co-rich matrix is mainly governed by nearest-neighbor effects. All energies are given in meV.

| Element | SQS 1 | | SQS 2 | | SQS 3 | |
| --- | --- | --- | --- | --- | --- | --- |
| | $E_{\text{Co-rich},X}$ | $E_{\text{Co-rich},X,\text{nn}}$ | $E_{\text{Co-rich},X}$ | $E_{\text{Co-rich},X,\text{nn}}$ | $E_{\text{Co-rich},X}$ | $E_{\text{Co-rich},X,\text{nn}}$ |
| Al | 250.8 | $-25.8$ | 305.7 | 29.1 | 293.5 | 16.9 |
| Ti | 272.8 | $-36.3$ | 338.8 | 29.7 | 352.2 | 43.2 |
| Nb | 284.9 | $-56.9$ | 325.2 | $-16.6$ | 372.8 | 30.9 |
| Ta | 299.8 | $-51.6$ | 344.9 | $-6.5$ | 390.5 | 39.1 |

In summary, the effect of a Co-rich matrix is explained mainly by the nearest-neighbor interaction. An increase in the Co concentration can therefore be expected to increase the energy of the alloying element in the nickel matrix so that the respective phase is stabilized and its solvus temperature increased.

### 3.2. Influence of Cr

### 3.2.1. Cr-Cr Interaction at Low Concentrations

Experimentally, it is known that Cr preferably bonds to unlike atoms [19]. However, a calculation of the Cr-Cr nearest-neighbor interaction energy in a Nickel matrix, performed identically to that in Section 3.1.1, yields an interaction energy of $-50$ meV when the Cr atoms are initialized with opposite magnetic moments to each other, rendering this configuration favorable. (If the Cr atoms are initialized with magnetic moments parallel to each other and to that of the Ni atoms, the final state has identical moments on both atoms and an interaction energy of 27 meV.)

There are several possible explanations for Cr atoms not favoring nearest-neighbor positions: In alloys with high Cr-content, the material is not ferromagnetic, so magnetic effects may change the nearest-neighbor interactions. Other configurations, for example next-nearest neighbor positions, may be more favorable as suggested by the existence of a $\text{Ni}_3\text{Cr}$ $\text{DO}_{22}$ phase [20]. Furthermore, at higher concentrations of Cr, more complicated interactions between the Cr atoms may render nearest-neighbor positions unfavorable.

The next-nearest-neighbor position is indeed even more favorable in a Ni matrix with an interaction energy of $-122$ meV when the Cr atoms have opposite magnetic moments to each other (the resulting state has moments of $\pm 1.59 \, \mu_B$). A ferromagnetic final state (with magnetic moments of $+1.18 \, \mu_B$) leads to an interaction energy of $-52$ meV; a state where both Cr atoms have the same magnetic moment, but opposite to the Ni matrix (with magnetic moments of $-1.66 \, \mu_B$) which has an interaction energy of $-67$ meV. The differences between these three magnetic states also suggest that magnetic interactions play an important role.

To further study the magnetic effects, non-spin polarized calculations were performed for the supercells $Ni_{108}$, $Ni_{107}Cr$, and $Ni_{106}Cr_2$. If the nearest-neighbor interaction is calculated in this state, the value changes drastically to 260 meV, making a nearest-neighbor position highly unfavorable.

In conclusion, the Cr-Cr interaction has been shown to be complex: In a ferromagnetic Ni matrix, the nearest-neighbor position is energetically favorable, but a next-nearest-neighbor position has an even lower energy and will thus be preferred. Furthermore, the interaction energy depends on the magnetic state of the Cr atoms. In an unmagnetic Ni matrix, Cr-Cr nearest-neighbor interactions are unfavorable.

### 3.2.2. Interaction between Cr and Other Alloying Elements at Low Concentration

The interaction energy between Cr and the other alloying elements was calculated in the same way as in Sections 3.1.1 and 3.2.1. In addition, non-spin-polarized calculations were also performed to see the effect of magnetic interactions. Results are shown in Table 3.

**Table 3.** Nearest-neighbor interaction energy (in meV) between Cr and other alloying atoms calculated in a matrix with 108 atoms. Results for both spin-polarized and non-polarized calculations are shown. Additionally shown are the atomic volumes (in units $\text{Å}^3$) of the elements in a (spin-polarized) $Ni_{107}X$ cell, calculated by a Voronoi construction.

| Element | Magnetic | Nonmagnetic | Volume |
|---|---|---|---|
| Al | 166 | 270 | 11.109 |
| Ti | 181 | 371 | 11.264 |
| Co | 32 | −35 | 10.795 |
| Nb | 172 | 435 | 11.541 |
| Ta | 186 | 446 | 11.557 |

In both cases, the interaction between Cr and Co is small. For the other elements, the energies in the spin-polarized calculation are close to each other. However, the magnetic state is not the same in all cases: For Al and Ti, an anti-ferromagnetic moment of Cr is more favorable, whereas the energetically lowest state for Cr-Nb and Cr-Ta is ferromagnetic with the anti-ferromagnetic polarization of Cr having an interaction energy that is larger by about 80 meV in both cases. It should be noted that converging to the correct magnetic state is difficult in these cases and requires the correct initialization of the magnetic moments and correct choice of the minimization algorithm. The calculation may converge to a state with a large ferromagnetic, large anti-ferromagnetic, or very small moment on the Cr atom, so it seems that there are actually three local minima for the electronic state.

In the non-spin-polarized case, the interaction energy is generally higher. In this case, there is a good correlation between the interaction energy and the atomic volume of the alloying elements (calculated using a Voronoi construction), showing that the interaction energy is dominated by the elastic repulsion between large ions. In the magnetic case, the increased elastic energy is countered by lowering the energy due to the ferromagnetic Cr orientation when placed next to Nb and Ta.

### 3.2.3. Construction of $Ni_{26}Cr_6$ Cells

To check the effect of large Cr concentrations, cells with composition $Ni_{26}Cr_6$, corresponding to a concentration of 18.75 at.-%, were created in two different ways.

The results from Section 3.2.1 suggest that Cr-Cr nearest-neighbor positions are less favorable than next-nearest-neighbor positions, and there is also experimental evidence that Cr atoms avoid these positions. (This is also corroborated by the fact that the energies of the SQS cells with Cr-Cr nearest-neighbor bonds that are discussed below are considerably larger.) Therefore, $Ni_{26}Cr_6$ cells with no Cr atoms in nearest-neighbor positions were created using the software SOD [18]. The seven possible configurations are shown in Figure 2. To ascertain the most favorable magnetic state of the cells, initial static calculations for each of the cells were performed where the magnetic moments of the Cr atoms were

varied systematically. Afterwards, the most favorable magnetic state was used to initialize a calculation with relaxation of ions and cell size.

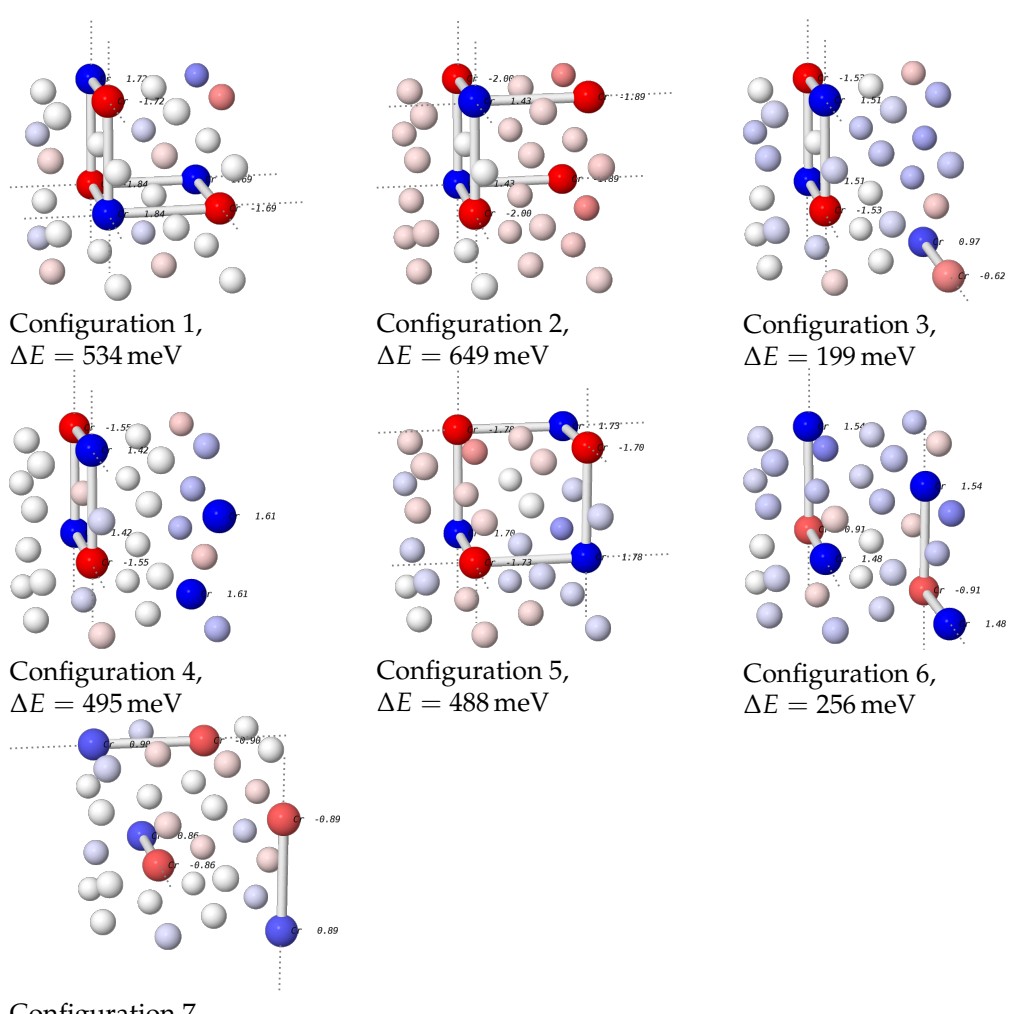

Configuration 1,
$\Delta E = 534$ meV

Configuration 2,
$\Delta E = 649$ meV

Configuration 3,
$\Delta E = 199$ meV

Configuration 4,
$\Delta E = 495$ meV

Configuration 5,
$\Delta E = 488$ meV

Configuration 6,
$\Delta E = 256$ meV

Configuration 7,
$\Delta E = 0$ meV

**Figure 2.** The seven possible $Ni_{26}Cr_6$ supercells with no nearest-neighbor bonds between Cr atoms. Cr atoms are plotted with a larger radius. Next-nearest-neighbor bonds inside the cell are shown, periodic nearest-neighbor bonds are depicted as dashed lines. The color of the atoms encodes the magnetic moments in the final state. Magnetic moments for the Cr atoms are listed in the plots. For all configurations, energy differences relative to the lowest-energy configuration 7 are stated.

The most favorable configuration was configuration 7 where the Cr atoms form three independent chains when periodic boundary conditions are taken into account. Energies of the other structures were larger by energies of up to 649 meV.

In the most favorable structure, the Cr atoms in each chain have opposing magnetic moments, whereas the moments in the Ni matrix are small. To check whether this structure is most favorable due to this magnetic interaction, we performed another calculation with no spin polarization. The energy of this was larger by only 84 meV, considerably less than the energy difference to most of the other structures.

It is therefore most plausible that the energy of the $Ni_{26}Cr_6$ cell is mainly determined by elastic interactions between the Cr atoms. This is corroborated further by the fact that structures 1, 2 and 5, which have an $L1_2$-like structure with two Cr atoms replaced by Ni, have the highest energies. Since the ionic radius of Cr is larger than that of Ni, the Ni-Ni-bonds in positions where a Cr atom is "missing" are probably highly strained and thus increase the energy of the structure. The only exception to this is configuration 4 which is similar to configuration 3, but has a considerably higher energy.

We can also use these structures to further assert that Cr-Cr nearest-neighbor bonds are unfavorable. To do so, one Cr atom in configuration 7 was shifted into a nearest-neighbor position to the other atom on its chain while leaving the number of Cr-Cr atoms on next-nearest-neighbor positions unchanged. The energy difference between these two configurations is 328 meV. This number is of the same order of magnitude as the nonmagnetic nearest-neighbor interaction energy calculated in Section 3.2.1, showing that nearest-neighbor bonds between Cr atoms are indeed unfavorable.

In addition, we created two cells with a special quasirandom structure (SQS) with a composition of $Ni_{25}Cr_6X$ to be used in the calculations of the alloying element interactions [17]. Replacing the X atom with Ni leads to a $Ni_{26}Cr_6$ cell that serves as reference state for the alloying calculations. Using this procedure has the advantage that in calculating the alloying effects, two cells are compared that only differ by the replacement of a single Ni atom with an alloying atom, so that interactions between the Cr atoms remain the same. Figure 3 shows the SQS cells.

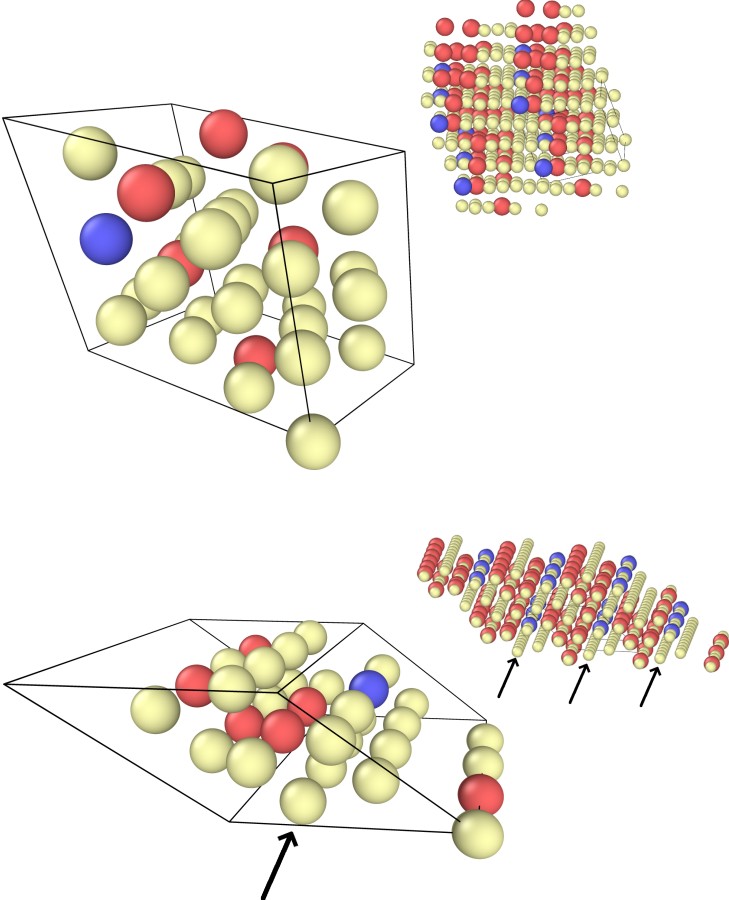

**Figure 3.** The two special quasirandom structures with composition $Ni_{25}Cr_6X$; Ni atoms in yellow, Cr atoms in red, alloying element in blue. The smaller pictures show a periodically repeated pattern, arrows in the second picture mark close-packed planes that do not contain Cr atoms.

The energies of the two $Ni_{26}Cr_6$ SQS cells (with a Ni atom in place of the alloying atom) are considerably larger than that of the most favorable state from Figure 2 by 1378 meV and 1391 meV, further confirming that Cr-Cr nearest-neighbor bonds are unfavorable. In both cells, 64 initializations of the magnetic moments of the Cr atoms were attempted to find the best magnetic state. In the first cell, three Cr atoms have positive magnetic moments, the other three have negative moments, and the magnetic moments on the nickel atoms are low overall, so that the net magnetic moment of the cell is small. In the second configuration, there are four Cr atoms with positive moments, leading to larger moments

on the Ni atoms and a larger overall magnetic moment. It may seem surprising that the energies of both cells are very close despite magnetic interactions being larger in SQS 2. This is probably due to a cancellation of elastic and magnetic energies as will be discussed in the next section.

### 3.2.4. Interaction in Cells with High Cr Content

Interaction effects were calculated for $Ni_{25}Cr_6X$ in the two SQS cells from Figure 3 and in the lowest energy cell without nearest neighbors, configuration 7 from Figure 2. There are three geometrically different positions in this cell as calculated using SOD [18], with 2, 3, and 4 nearest-Cr-neighbors of the alloying element, see Figure 4. These cells are denoted as "non-nn-cells" in the following. In all cases, different initial magnetizations for the alloying element and the Cr atoms were used to find the best final state.

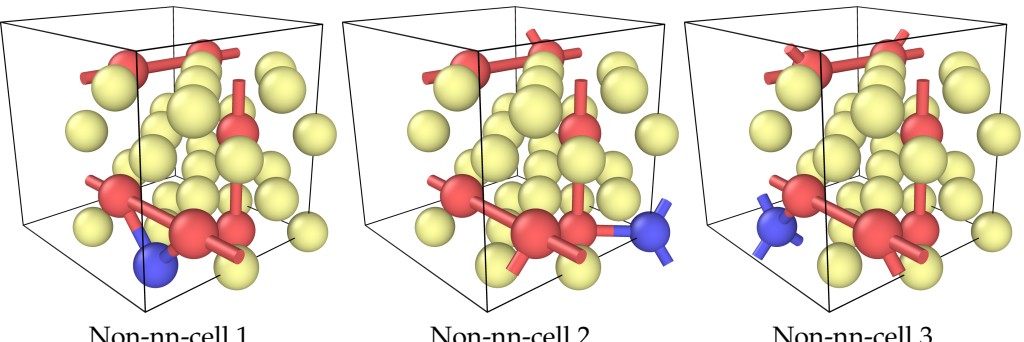

Non-nn-cell 1      Non-nn-cell 2      Non-nn-cell 3

**Figure 4.** The three geometrically distinct positions of an alloying atom in the energetically best cell $Ni_{25}Cr_6X$ from Figure 2. Nearest-neighbor bonds between the alloying atom and Cr and next-nearest-neighbor bonds between Cr atoms are shown. Ni atoms in yellow, Cr atoms in red, alloying element in blue.

The energy to transfer an alloying element from a pure Ni-matrix to a Cr-rich matrix is calculated in the same way as in Equation (2):

$$E_{\text{Cr-rich},X} = E(Ni_{108}) + E(Ni_{25}Cr_6X) - (E(Ni_{107}X) + E(Ni_{26}Cr_6)). \qquad (4)$$

A negative number means that placement in the Cr-rich matrix is more favorable, a positive number means that it is less favorable.

Table 4 shows the calculated transfer energy which is positive for all elements. This shows that a Cr-rich matrix will tend to displace the studied elements to other phases compared to a Cr-free Ni matrix. The absolute values vary strongly, but it should be noted that the energy of SQS 1 and non-nn-cell 1 are always rather close. Both these cells have two Cr-neighbors on the alloying atom. The transfer energy of SQS 2—which also has two nearest-Cr-neighbors—is consistently lower than that of SQS 1 by 120 meV–160 meV; reasons for this will be discussed below.

Apart from Co, which will be discussed separately, it is apparent that the transfer energy increases significantly with the number of nearest-Cr-neighbors of the alloying atom. This suggests that nearest-neighbor effects play some role in determining the energy of the alloying atom. To account for this, we can subtract the nearest-neighbor interaction energy as in Equation (3) and calculate

$$E_{\text{Cr-rich},X,\text{nn}} = E_{\text{Cr-rich},X} - N_{\text{nn}}E_{\text{nn}}, \qquad (5)$$

where $N_{\text{nn}}$ is the number of nearest-Cr-neighbors of the alloying atom and $E_{\text{nn}}$ is the nearest-neighbor interaction energy from Table 3. Using the value of $E_{\text{nn}}$ calculated from a magnetic matrix results in numbers that differ strongly for the three non-nn-cells.

**Table 4.** The transfer energy for an element between a pure nickel matrix and a $Ni_{25}Cr_6X$ cell. The first sub-table shows the transfer energy $E_{Cr-rich,X}$ itself. The second sub-table shows the number of nearest neighbours in each cell and the energy $E_{Cr-rich,X,\,nn}$ that accounts for nearest-neighbor effects, using the magnetic interaction energy from Table 3. The third sub-table shows $E_{Cr-rich,X,\,nn}$ when using the non-magnetic interaction energy. Detailed explanation in the text. All energies are given in meV.

| | $E_{Cr-rich,X}$ | | | | |
| | non-nn-cell-1 | non-nn-cell-2 | non-nn-cell-3 | SQS-1 | SQS-2 |
|---|---|---|---|---|---|
| Al | 252 | 519 | 744 | 332 | 192 |
| Ti | 374 | 715 | 1243 | 384 | 211 |
| Co | 210 | 224 | 236 | 164 | 223 |
| Nb | 324 | 688 | 1284 | 297 | 132 |
| Ta | 368 | 728 | 1321 | 337 | 164 |

| | $E_{Co-rich,X,nn}$ with Magnetic nn Interaction | | | | |
| | non-nn-cell-1 | non-nn-cell-2 | non-nn-cell-3 | SQS-1 | SQS-2 |
|---|---|---|---|---|---|
| **Number Nearest Neighbors** | **2** | **3** | **4** | **2** | **2** |
| Al | −81 | 20 | 78 | 0 | −140 |
| Ti | 12 | 173 | 519 | 22 | −150 |
| Co | 146 | 128 | 109 | 100 | 160 |
| Nb | −20 | 172 | 595 | −47 | −212 |
| Ta | −4 | 169 | 577 | −35 | −208 |

| | $E_{Co-rich,X,nn}$ with Non-Magnetic nn Interaction | | | | |
| | non-nn-cell-1 | non-nn-cell-2 | non-nn-cell-3 | SQS-1 | SQS-2 |
|---|---|---|---|---|---|
| Al | −289 | −293 | −338 | −209 | −349 |
| Ti | −369 | −399 | −243 | −359 | −531 |
| Co | 280 | 328 | 376 | 233 | 293 |
| Nb | −547 | −618 | −458 | −574 | −739 |
| Ta | −524 | −612 | −464 | −555 | −729 |

However, because the $Ni_{25}Cr_6X$ cells have rather low magnetic moments and the energy difference between non-nn-cell 1 and 2 is close to the nonmagnetic nearest-neighbour interaction energy, we can use the nonmagnetic interaction energy instead. Doing this results in more consistent values for the elements Al, Ti, Nb, and Ta. All values are negative, showing that the energy of the $Ni_{25}Cr_6X$ cell is lower than expected from a purely nonmagnetic nearest-neighbor repulsion.

As mentioned above, the transfer energy of $Ni_{25}Cr_6X$ is consistently lower in SQS 2 than in SQS 1 by about 120–160 meV. (This is also true for the total energy of these cells because the energy of $Ni_{26}Cr_6$ SQS 1 and SQS 2 is almost the same.) A closer look at the cells (Figure 3) shows that SQS 2 contains a plane of Ni atoms next to a chain of Cr atoms. This structure can be expected to be under tensile strain. Due to these strains, this plane can accomodate an alloying atom of larger radius more easily than SQS 1. However, since the overall magnetic moment of all SQS 2 cells is larger than that of SQS 1, magnetic effects may also lower the energy of SQS 2 relative to SQS 1. To discern these effects, we performed non-spin-polarized calculations for the two SQS cells. In the non-polarized cells, the total energy of $Ni_{25}Cr_6X$ is lower for SQS 2 than SQS 1 for the elements Ti, Nb, and Ta, but almost identical for Al, and the difference is smaller than before (between 100 meV and 130 meV). Thus, both magnetic and elastic effects are responsible for the lower energy of the SQS 2 $Ni_{25}Cr_6X$ cells. That the $Ni_{26}Cr_6$ SQS 2 is very close in energy to SQS 1 is due to the cancellation of two effects: the large elastic strains in the Ni planes increase the energy, but the stronger magnetic interaction lowers it.

In summary, the results show that the effect of a high Cr content on the solution energy of Al, Ti, Nb, and Ta is complex. The nearest-neighbor repulsion is larger in a situation

without spin polarization. Since increasing the Cr content reduces the magnetic effects, it can be expected that the repulsion between Cr and the alloying elements increases as well. Since the repulsion in a non spin-polarized calculation correlates well with the atomic volume, the energy seems to be dominated by elastic effects. The final energy is lower than predicted by the purely non-magnetic interaction.

For Co, the transfer energy $E_{Cr-rich,Co}$ is about 200 meV for all configurations, independent of the number of nearest neighbors. Taking the nearest-neighbor interaction energy into account changes the transfer energy to a range between 100 meV and 160 meV. As the ionic radius of Co is very close to that of Ni, elastic effects can be expected to be small. To test this, some calculations without relaxation were performed, confirming that ionic relaxation due to Co is small.

On the other hand, Co has a large magnetic moment when placed in a Ni matrix, whereas the Cr-rich cells have low magnetic moments. This suggests that the effect of a Cr-rich matrix is due to magnetic interactions. To check this, we calculated the energy difference between a Co atom in a magnetic Ni matrix and compared it to that in a non-magnetic Ni matrix. This difference is 310 meV, about 100 meV larger than $E_{Cr-rich,Co}$ and close to the values of $E_{Cr-rich,Co,nn}$ when using the nonmagnetic interaction energy. Some deviations are to be expected because Co does retain some magnetic moment in the $Ni_{25}Cr_6Co$ cell.

To further confirm this picture, we simulated the two SQS structures without spin polarization. In this case, $E_{Cr-rich,Co}$ is rather small with values of 72 meV and 59 meV. Taking the nonmagnetic nearest-neighbor interaction into account to calculate $E_{Cr-rich,Co,nn}$ for the nonmagnetic case results in very small values of 2 meV and 10 meV. Thus, in the non-magnetic case, the energy is predicted very well.

In conclusion, Co has a higher energy in a Cr-rich matrix because the magnetic interactions, which lower the energy of Co in pure Ni, are disturbed, whereas elastic effects do not play a significant role.

## 4. Discussion

It was shown in the previous section that both Co and Cr increase the energy of the phase-forming elements (Al, Ti, Nb, Ta) in the nickel matrix and thus may stabilize the respective phases. The effect of Co is mainly due to the direct interaction energy between Co and an alloying element in the nearest-neighbor position. For Cr, interaction energies are generally higher, especially if nonmagnetic nearest-neighbor interactions are considered as is appropriate for high Cr content. In cells where the Cr content is large, the energy is not as large as expected by the direct nearest-neighbor interaction.

From this it can be concluded that the solvus temperature of the phases considered should increase with the Co and Cr content. Since the effect is simply cumulative for Co (the number of nearest-neighbor bonds of an alloying atom directly determine the energy increase), it can be expected that this increase should be almost linear.

For Cr, on the other hand, interaction energies are larger so it should be expected that the slope of the solvus line is larger than for Co. Furthermore, since the increase in energy is not directly proportional to the number of bonds and is lower than expected at high concentrations (if nonmagnetic interaction energies are considered), the slope of the line should decrease with increasing concentration.

To confirm this, simple thermodynamical calculations with the software Thermocalc [21] using the TTNI8 database were performed. We calculated pseudobinary phase diagrams, excluding all phases except for fcc, liquid, and the phase under consideration. The basic alloy composition was chosen as Ni 5 at-% Al for $\gamma'$, Ni 8 at-% Nb for $\delta$ and Ni 8 at-% Ti for $\eta$. Phase boundaries were calculated for a Co or Cr content up to 30 at-%.

The results of the calculation are shown in Figure 5. The solvus lines behave as expected, with a lower slope and a more linear shape for Co and an initially higher slope and a more curved shape for Cr.

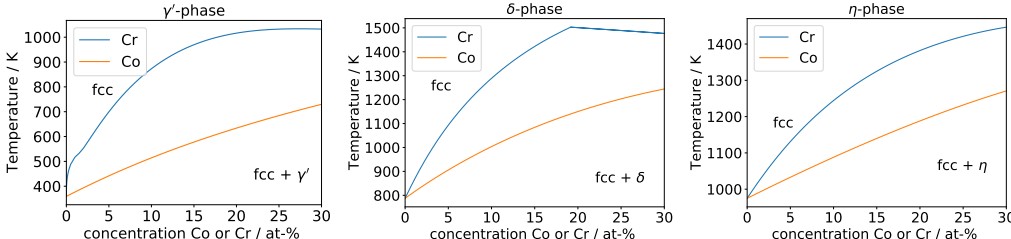

**Figure 5.** Pseudobinary phase diagrams showing the solvus lines of the phases $\gamma'$, $\delta$, $\eta$ as a function of the Cr or Co content. The alloy composition at 0% is Ni 5 at-% Al for $\gamma'$, Ni 8 at-% Nb for $\delta$ and Ni 8 at-% Ti for $\eta$. The kink in the line for the $\delta$-phase is due to the solvus line meeting the solidus line.

For the $\gamma'$-phase, it should be noted that Thermocalc predicts that Cr preferentially dissolves in $\gamma'$, so this solvus line does not capture a displacing effect, but rather a direct stabilization of the phase. For Co in $\gamma'$ and both for Co and Cr in $\delta$ and $\eta$, Thermocalc predicts low solubility, so the solvus line should mainly represent the energy increase of the element in a matrix rich in Co or Cr.

The conclusions drawn here from our simulations are also supported by experimental findings. Heslop [22] determined the $\gamma'$-solvus temperatures in the two model alloys Ni-20Cr-xTi-(x/2)Al and Ni-20Cr-20Co-xTi-(x/2)l. At 2% Ti (i.e., 1% Al) he found solvus temperatures of about 820 °C and 910 °C for the former and latter alloy, respectively. However, it should be noted that the difference diminished with increasing Ti-content and eventually turned around at about 5% Ti. This illustrates again that concentration-dependent effects may occur. Moreover, further interactions may take place in multi-component alloys not considered here. In the works of [3,23,24], a new Nb-containing wrought superalloy was developed. In this context, it was found that Co-addition increases the solvus temperature of the Nb-rich high temperature precipitates that were identified to consist of $\eta$- and $\delta$-phase [25]. Finet et al. [26] studied the formation of the $\delta$ and $\eta$-phase in two experimental alloys with varying Cr and Co content. In all cases, they observed an increase in the volume fraction of the phases with increased Cr content whereas an increased Co content further increased the volume fraction in one alloy, but not in the other.

The results of the DFT calculations also show that a high Cr content increases the energy of Co by about 200 meV. According to the research findings of [8,9], the energy needed to transfer Co from a pure nickel matrix to the $\gamma''$- or $\delta$-phase is approximately 300 meV. This energy is thus significantly reduced when Cr is present in the Ni matrix, so it can be expected that at least some Co will be found in the $\gamma''$- or $\delta$-phase.

## 5. Conclusions

In this paper, it is shown that a high content of either Co or Cr in a nickel matrix significantly changes the solution energy of relevant alloying elements. It is important to consider these effects when trying to predict the properties of nickel-base alloys from simulations.

In more detail, the following key results are obtained:

- The self-interaction between two Co atoms is small, allowing Co atoms to form random solid solutions.
- A high Co content increases the solution energy of the phase-forming elements considerably by 250–391 meV and can thus be expected to increase the solvus temperature of the respective phases.
- The increase in the solution energy can be explained by the nearest-neighbor interaction between the phase-forming elements and Co.
- The self-interaction of two isolated Cr atoms in a nickel matrix is attractive for nearest-neighbor positions and even more so for next-nearest-neighbor positions. However, in non-spin-polarized calculations, Cr atoms in nearest-neighbor positions strongly repel each other. The comparison between $Ni_{26}Cr_6$ cells with no Cr atoms in a nearest-neighbor position and randomly generated SQS cells shows that cells with Cr atoms

in nearest-neighbor positions have considerably larger energy. This is in agreement with experimental results that show that Cr atoms avoid nearest-neighbor positions.
- The solution energy of Al, Ti, Nb, and Ta strongly increases in a Cr-rich matrix. Again, this implies that a high Cr content increases the solvus temperature of the respective phases.
- The increase in the solution energy is not explained by the nearest-neighbor interaction energy between the phase-forming element and Cr in a nickel matrix. However, if interaction energies are calculated without spin polarization, a consistent picture emerges. The effect of a high Cr content on the solution energies is thus not only governed by the next-neighbor interaction, but also by the fact that Cr alters the magnetic state of the alloy.
- Placing a Co atom in a Cr-rich matrix increases the solution energy of the Co atom. This is not explained by the nearest-neighbor interaction, but by the fact that a Cr-rich matrix strongly reduces the magnetic moments of the Ni atoms, thus weakening the Ni-Co bonds.
- The predicted effects of a Co- or Cr-rich matrix are in agreement with experimental data and results from CALPHAD calculations.
- The results also stress the importance of accounting for magnetic effects when calculating the properties of nickel-base superalloys using DFT.

**Author Contributions:** Conceptualization, M.B. and J.R.; methodology, M.B.; validation, M.B.; writing—original draft preparation, M.B.; writing—review and editing, M.B. and J.R.; supervision, J.R.; project administration, M.B.; funding acquisition, M.B. and J.R. All authors have read and agreed to the published version of the manuscript.

**Funding:** This research was partially funded by the Deutsche Forschungsgemeinschaft grant DFG BA 1795/13-1. Computations were performed at the Hochleistungsrechenzentrum Norddeutschland under grant nic00029. We acknowledge support by the Open Access Publication Funds of Technische Universität Braunschweig.

**Institutional Review Board Statement:** Not applicable.

**Data Availability Statement:** Data available on request.

**Acknowledgments:** We would like to thank Martin Bergner for help in setting up the Thermocalc calculations and Tarakeshwar Lakshmipathy for performing some preliminary calculations on Cr-rich systems.

**Conflicts of Interest:** The authors declare no conflict of interest.

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
