# Peer review of "Effect of Co and Cr on the Stability of Strengthening Phases in Nickelbase Superalloys"

_crystals, doi:10.3390/cryst12081084_

Round 1
Reviewer 1 Report
-
If possible, it is desirable to have experimental or literature support for computational studies.
Author Response
"Please see the attachment"
Reviewer 2 Report
This manuscript reported Effect of Co and Cr on the stability of strengthening phases in nickelbase superalloys. As a whole, the article has a good innovative, the experimental contents are very rich and the analysis of the article is comprehensive and professional. However, some problems need to be addressed. I recommend to minor revision.
1. Some grammatical problems were found in manuscript, which need to be revised.
2. The abstract part needs to add some specific experimental data to reflect the innovation of the paper.
3. The introduction section should be improved; some recently published papers about nickelbase superalloy should be added and described for comparison. And what is your innovation?
4. Where is the conclusion?
5. All tables are irregular and have no clear boundary lines.
Author Response
"Please see the attachment."
Reviewer 3 Report
Dear authors,
The text is well expressed, but please, follow the indications made in the file attached.
After ameding the comments made, the manuscript should be suitable for publication. Please, take into account that a final point of conclusions is missing and must be added to the final version before publication.
Author Response
"Please see the attachment."